

# Evaluation of psychological stress in scientific researchers during the 2019–2020 COVID-19 outbreak in China

Xueyan Zhang[1,2,*], Xinyu Li[1,2,*], Zhenxin Liao[3], Mingyi Zhao[4] and Quan Zhuang[1,5]

[1] Transplantation Center, The 3rd Xiangya Hospital of Central South University, Changsha, Hunan, China
[2] Xiangya School of Medicine of Central South University, Changsha, Hunan, China
[3] Xiangya School of Public Health of Central South University, Changsha, Hunan, China
[4] Department of Pediatrics, The 3rd Xiangya Hospital of Central South University, Changsha, Hunan, China
[5] Research Center of National Health Ministry on Transplantation Medicine, Changsha, Hunan, China
* These authors contributed equally to this work.

## ABSTRACT

**Background:** Beginning in December 2019, coronavirus disease 2019 (COVID-19) caused an outbreak of infectious pneumonia. The Chinese government introduced a series of grounding measures to prevent the spread of COVID-19. The living and working patterns of many scientific researchers also underwent significant changes during this period.

**Methods:** An opportunity sample ($n$ = 251) was obtained in China using a questionnaire with 42 questions on scientific research progress and psychological stress during the COVID-19 epidemic.

**Results:** Of the 251 participants, 76.9% indicated that their research was affected by the COVID-19 outbreak, and participants who were affected by the outbreak had higher stress levels than those who were not affected. Participants who conducted COVID-19 research and indicated concern that they would fail to finish the research on time were more likely to indicate high levels of stress. Respondents indicated that extending deadlines (64.1%), receiving support from superiors for research (51.8%), and increasing benefits for researchers (51.0%) would likely relieve outbreak-related stress.

**Conclusion:** The COVID-19 outbreak had a major impact on the experiments of researchers in the life sciences, especially in basic and clinical medicine. It has also caused high levels of psychological stress in these populations. Measures should be taken to relieve psychological pressure on basic medical researchers and students who will soon complete their degrees (e.g., Master's and PhD candidates in graduation years).

Corresponding authors
Mingyi Zhao, 36163773@qq.com
Quan Zhuang,
zhuangquansteven@csu.edu.cn

## INTRODUCTION

The outbreak of coronavirus disease 2019 (COVID-19), caused by severe acute respiratory syndrome coronavirus 2 (SARS-CoV-2), originated in Wuhan, China and quickly spread from human to human in December 2019 (*Lai et al., 2020*). On 30 January 2020, the World Health Organization declared the COVID-19 epidemic a public health emergency of international concern. As of 28 February 2020, China has confirmed 77,150 new coronavirus infections and 2,592 deaths (*Pediatric Committee, Medical Association of Chinese People's Liberation Army and Editorial Committee of Chinese Journal of Contemporary Pediatrics, 2020*; *Martinez, 2020*). Because of the increasing number of confirmed cases and deaths, negative emotions continued to spread (*Zhou, 2020*). Previous studies (*Hull, 2005*; *Wu, Chan & Ma, 2005a*, *2005b*) suggest we must examine the extent of psychological stress associated with the current epidemic and focus attention on those people most vulnerable to this psychological stress (*Shigemura et al., 2020*). Recent studies have focused on the psychological stress of the medical staff involved in epidemic prevention in China (*Xiao et al., 2020*). However, few studies have examined the impact of severe infectious disease outbreaks on the psychological state of researchers. The rise of stressors and strains in academic life has been widely reported (*Kinman, 2001*). Heavy workload and time and resource constraints have been highlighted as major work stressors in researchers. The work-home imbalance and role conflict and overload also have potential impact on academic stress level (*Gmelch, Lovrich & Wilke, 1984*; *Kinman, 2008*; *Tytherleigh et al., 2005*). At the same time, stress from dissatisfaction with pay and benefits has been reported (*Tytherleigh et al., 2005*). Management and leadership styles, a pressured higher education climate, and unhealthy competition also cause harmful stress (*Wellcome, 2020*).

To avoid further transmission of COVID-19, many industries were forced to shut down temporarily, and scientific and social research and education activities were paused in China (*Couzin-Frankel, 2020*). Furthermore, animal centers and practical labs were closed, and many scientific and social congresses and symposiums were cancelled, leaving postgraduates and scientific workers confined to their homes (*Service, 2020*). Therefore, many researchers' experimental progress was hindered (e.g., due to loss of samples and funds) (*Service, 2020*; *Tencent, 2020a*, *2020b*), which undoubtedly increased the psychological stress on academic and research staff. In addition, the stagnation of science education activities may cause an increase in students' graduation pressure, and even the delay of graduation (*Tencent, 2020a*).

In the current research, we propose the following hypotheses: (1) the COVID-19 outbreak aggravated psychological stress in researchers; (2) the stress levels and stressors in diverse populations would be different; (3) the demands for reducing stress in diverse populations would be different. We included 42 related questions in a questionnaire to test the above hypotheses. Respondents were categorized by research field, research degree and affiliation, etc.

## MATERIALS AND METHODS

### Study participants

A questionnaire was distributed to researchers in China to recruit an opportunity sample, and all respondents were asked to answer each question on their own. The targets of the questionnaire were identified as "scientific researchers", which requires the respondents to be involved in at least one research project in past 12 months. Questionnaires were distributed to research institution staff, university researchers and students participating in the research. They were all researchers with a confirmed scientific experience or people the authors had collaborated with before. Some respondents passed on the questionnaire to other qualified people to fill in. A total of 251 questionnaires was received after screening. As the investigators are scientific researchers, they have a high level of education and are familiar with the questions. In addition, the respondents' answers were internally consistent. In particular, the 193 respondents who chose 'At a standstill' or 'still in progress but slower than before' due to the epidemic situation (question 37) responded with psychological states (questions 38–39) that matched this answer. All participants provided written informed consent, and subjects were anonymous. It is not required by our institution to obtain ethical approval for a survey with a nonclinical sample and anonymised data, but we did obtain retrospective approval for the study protocol from the institutional review board (Ethics Committee) of the 3rd Xiangya Hospital, Central South University (20005-IRB).

### Questionnaire

We included 42 related questions in the questionnaire to acquire a comprehensive understanding of the progress of research projects and the current psychological stress level of researchers. The survey consisted of 24 questions assessing the subject's psychological stress (i.e., stress scale). The questionnaire incorporated modified questions from the stress response questionnaire (SRQ) and the Pittsburgh sleep quality index scale (PSQI) (*Pilz et al., 2018*) and considered the current COVID-19 epidemic (i.e., emotional state, somatic responses, sleep quality and behavior). The stress scale consisted of five self-evaluation options: (1) not at all, (2) occasionally, (3) sometimes, (4) often and (5) always. A score of 5 represented the highest level of stress.

We also assessed participants' research areas (e.g., whether they conduct research related to the novel coronavirus) and potential stagnation of research projects, including questions rated to (1) delay in scientific research projects, (2) sample or funding losses due to the current epidemic and (3) disruption of academic exchange activities. At the conclusion of the questionnaire, subjects were invited to evaluate some recommendations and proposed solutions for potential changes to scientific research in China, including extending deadlines for project completion, providing partial financial subsidies for scientific research losses, assigning professional personnel to guide and support scientific research projects, and prioritizing the return of researchers to work (see Supplement 1).

**Table 1 Demographic characteristics of respondents.**

| Demographic characteristics | | N | % |
|---|---|---|---|
| Gender | Male | 104 | 41.43 |
| | Female | 147 | 58.57 |
| Age | 18–24 | 109 | 43.43 |
| | 25–39 | 119 | 47.41 |
| | 40–59 | 22 | 8.76 |
| | ≥60 | 1 | 0.4 |
| Category of school or institution | Top universities | 177 | 70.52 |
| | General college | 13 | 5.18 |
| | Independent research institutes (including research institutes) | 4 | 1.59 |
| | University affiliated hospital | 55 | 21.91 |
| | Non-university affiliated hospital | 2 | 0.8 |
| Education background | Undergraduates | 79 | 31.47 |
| | Master candidate (non-graduate) | 28 | 11.16 |
| | Master candidate (graduation grade) | 9 | 3.59 |
| | PhD candidate (non-graduate year) | 23 | 9.16 |
| | PhD candidate (graduation year) | 20 | 7.97 |
| | Basic research staff (including postdoctoral) | 31 | 12.35 |
| | Clinical medical staff (including postdoctoral) | 61 | 24.3 |
| Title of technical post | Professor (researcher, chief physician) | 11 | 4.38 |
| | Associate professor (associate researcher, associate chief physician) | 22 | 8.76 |
| | Lecturer (assistant researcher, attending physician) | 44 | 17.53 |
| | None | 174 | 69.32 |
| Total | | 251 | 100 |

## Statistical analysis

Questionnaire results were summarized from the imported Excel file and analyzed using SPSS version 18.0 software (IBM Corp., Armonk, NY, USA). Quantitative variables were expressed as an average with a standard deviation (SD). Qualitative variables were expressed as numbers and percentages. Chi-squared ($\chi^2$) tests and analysis of variance (ANOVA) tests were used to compare psychological factors across social roles and age groups. A $p$ value less than or equal to 0.05 was considered statistically significant.

## RESULTS

### Participant demographics

Participants included scholars in the fields of life science (e.g., medicine, biology), engineering science (e.g., mechanical engineering, physiology, chemistry) and humanities and social sciences (e.g., law, literature). The gender ratio of the respondents was approximately 1:1. The average age of participants was 28.91 ± 8.65, most of whom were from colleges or university affiliated hospitals (Table 1). Participants consisted of seven groups of people: undergraduate students, Master's degree candidates (non-graduation

**Table 2 Scores and statistical analysis of researchers' stress levels.**

| Question number | Dimensions | n = 251 | | | |
|---|---|---|---|---|---|
| | | Min | Max | Means ± SDs | Median |
| 13–20 | Emotional state | 8 | 40 | 15.85 ± 7.93 | 14 |
| 21–25 | Somatic responses | 5 | 25 | 10.18 ± 5.05 | 9 |
| 26–29 | Sleep quality | 4 | 20 | 8.24 ± 4.08 | 7 |
| 30–36 | Behavior | 7 | 35 | 12.73 ± 5.50 | 11 |
| Total scores | | 24 | 120 | 46.99 ± 20.84 | 43 |

Note:
"±" Represents standard deviation.

**Table 3 Impact of the delay of project on researchers' stress levels.**

| Question number | Dimensions | Completion date of scientific research project was delayed by coronavirus | | F | p |
|---|---|---|---|---|---|
| | | Disagree (n = 83) | Agree (n = 168) | | |
| 13–20 | Emotional state | 13.72 ± 6.24 | 16.90 ± 8.47 | 9.194 | 0.003** |
| 21–25 | Somatic responses | 9.16 ± 4.21 | 10.68 ± 5.35 | 5.181 | 0.024* |
| 26–29 | Sleep quality | 7.59 ± 3.47 | 8.55 ± 4.32 | 3.131 | 0.078 |
| 30–36 | Behavior | 11.35 ± 4.42 | 13.41 ± 5.86 | 8.021 | 0.005** |

Notes:
* $p < 0.05$.
** $p < 0.01$.
"±" Represents standard deviation.

year), Master's degree candidates (graduation year), PhD candidates (non-graduation year), PhD candidates (graduation year), basic research staff (including postdoctoral), and clinical medical staff (including postdoctoral). Many participants were undergraduates and clinical medical staff without advanced degrees, who comprise the majority of researchers in China and are therefore the most vulnerable to research-related psychological stress from infectious disease outbreaks.

## Impact of epidemic-related scientific delays on stress levels

Of the 251 researchers surveyed, the average score of the population's stress level was 46.99 ± 20.84 points (full mark: 120 points). The median score was 43 points, the lowest score 24 points, and the highest score 120 points (Table 2). Participants whose progress was affected by the outbreak had higher levels of stress than participants who were not affected by the outbreak. Participants who indicated that they were affected by the epidemic expressed higher stress in emotional states, somatic responses, and behavior than participants who indicated they were not affected by the epidemic (Table 3).

We identified 14 possible predictors of high stress in researchers during the COVID-19 outbreak and conducted the variables with means and SDs on stress levels. As a result of the outbreak, researchers who were required to change or reduce experimental projects indicated they were under more pressure than those who did not have to change or reduce their experimental project. In addition, researchers were affected by peer pressure because their colleagues had been reporting on new coronavirus-related research (Table 4).

**Table 4  All the variables with means and SDs on stress levels.**

| Variable | Answers | Numbers | Means ± SDs | t | p |
|---|---|---|---|---|---|
| Delayed completion date of research project. | Disagree | 83 | 41.82 ± 16.41 | −3.102 | 0.002** |
| | Agree | 168 | 49.55 ± 22.32 | | |
| The blocked research projects caused by the COVID-19 has influenced the graduation/project conclusion/funds applications | Disagree | 122 | 42.93 ± 17.68 | −3.071 | 0.002** |
| | Agree | 129 | 50.83 ± 22.85 | | |
| Original research programs need to be changed or cut | Disagree | 168 | 43.20 ± 16.65 | −3.681 | 0.000** |
| | Agree | 83 | 54.67 ± 25.89 | | |
| The epidemic reduced the timeliness of the experimental results. | Disagree | 157 | 44.33 ± 18.43 | −2.484 | 0.014* |
| | Agree | 94 | 51.44 ± 23.79 | | |
| The epidemic has caused the loss of experimental materials and samples. | Disagree | 169 | 44.27 ± 17.47 | −2.652 | 0.009** |
| | Agree | 82 | 52.60 ± 25.69 | | |
| The COVID-19 asked more energy input | Disagree | 91 | 43.37 ± 17.19 | −2.245 | 0.026* |
| | Agree | 160 | 49.05 ± 22.45 | | |
| The COVID-19 has already, or is about to, caused a great loss for projects | Disagree | 160 | 44.63 ± 18.68 | −2.252 | 0.026* |
| | Agree | 91 | 51.14 ± 23.72 | | |
| The COVID-19 has influenced the original academic exchange activities | Disagree | 82 | 44.96 ± 18.50 | −1.075 | 0.284 |
| | Agree | 169 | 47.98 ± 21.87 | | |
| The COVID-19 has Influenced the cooperation with other organizations | Disagree | 123 | 45.14 ± 19.30 | −1.384 | 0.168 |
| | Agree | 128 | 48.77 ± 22.15 | | |
| Colleagues have carried out research projects on COVID-19. | Disagree | 150 | 45.34 ± 19.11 | −1.535 | 0.126 |
| | Agree | 101 | 49.45 ± 23.05 | | |
| I am doing/have done a research project on COVID-19. | Disagree | 197 | 46.16 ± 19.42 | −1.038 | 0.303 |
| | Agree | 54 | 50.02 ± 25.32 | | |
| Feeling great pressure from colleagues who conduct projects on COVID-19 | Disagree | 184 | 44.06 ± 17.55 | −3.153 | 0.002** |
| | Agree | 67 | 55.04 ±2 6.48 | | |
| The block of the scientific researched has caused the adverse effects on your career | Disagree | 151 | 44.11 ± 17.80 | −2.570 | 0.011* |
| | Agree | 100 | 51.35 ± 24.18 | | |
| The block of the scientific researched has caused the adverse effects on your salaries | Disagree | 168 | 45.15 ± 19.30 | −1.881 | 0.062 |
| | Agree | 83 | 50.72 ± 23.34 | | |
| Total | | 251 | | | |

Notes:
* $p < 0.05$.
** $p < 0.01$.

We did not report a separate analysis of correlation between research stress and influencing factors in different research disciplines, due to the limited sample size. The impact of "Original research programs need to be changed or cut" on stress levels and the impact of "Feeling great pressure on colleagues who conduct projects on COVID-19" on stress levels are additionally listed in Tables 5 and 6.

## Responses regarding recommendations to improve conditions for scientific researchers

Nine proposed solutions were considered by researchers to possibly ease their stress (Fig 1). The top recommendation was prolonging the graduation/project conclusion/the

**Table 5 The impact of "Original research programs need to be changed or cut" on stress levels.**

| Question number | Dimensions | Original research programs need to be changed or cut | | F | p |
| --- | --- | --- | --- | --- | --- |
| | | Disagree ($n = 168$) | Agree ($n = 83$) | | |
| 13–20 | Emotional state | 14.25 ± 6.31 | 19.08 ± 9.74 | 22.391 | 0.000** |
| 21–25 | Somatic responses | 9.32 ± 4.14 | 11.93 ± 6.16 | 15.775 | 0.000** |
| 26–29 | Sleep quality | 7.80 ± 3.51 | 9.12 ± 4.94 | 5.971 | 0.015* |
| 30–36 | Behavior | 11.83 ± 4.62 | 14.54 ± 6.61 | 14.184 | 0.000** |
| Total scores | | 43.20 ± 16.65 | 54.67 ± 25.89 | 18.001 | 0.000** |

Notes:
* $p < 0.05$.
** $p < 0.01$.

**Table 6 The impact of "Feeling great pressure from colleagues who conduct projects on COVID-19" on stress levels.**

| Question number | Dimensions | Feeling great pressure on colleagues who conduct projects on COVID-19 | | F | p |
| --- | --- | --- | --- | --- | --- |
| | | Disagree ($n = 168$) | Agree ($n = 83$) | | |
| 13–20 | Emotional state | 14.80 ± 6.82 | 18.73 ± 9.89 | 12.628 | 0.000** |
| 21–25 | Somatic responses | 9.57 ± 4.32 | 11.87 ± 6.38 | 10.605 | 0.001** |
| 26–29 | Sleep quality | 7.78 ± 3.48 | 9.49 ± 5.21 | 8.982 | 0.003** |
| 30–36 | Behavior | 11.92 ± 4.74 | 14.96 ± 6.75 | 15.861 | 0.000** |
| Total scores | | 44.06 ± 17.55 | 55.04 ± 26.48 | 14.378 | 0.000** |

Notes:
* $p < 0.05$.
** $p < 0.01$.

deadline of the application of the funds, with 161 of 251 (64.14%) respondents regarding it effective. Receiving support from superiors (51.79%) and improving the welfare of researchers (51.00%) came next. Academic cooperation (27.49%) and meetings (21.91%) received lower levels of endorsement. These demands varied statistically between clinical staff and basic medical researchers, as well as between master's and doctoral students (Tables 7 and 8).

## COVID-19 affected research progress differently across research fields and seniority

As a result of the COVID-19 outbreak, 47.11% of researchers in the field of science indicated their research programs were halted, and 32.00% of researchers indicated their programs, while ongoing, were slower than before the epidemic began. However, the COVID-19 epidemic has had relatively little impact on researchers in the field of humanities, with most social science researchers indicating a slower pace of research (6 out of 12) or a lack of impact of COVID-19 on their research (5 out of 12). It is worth noting that we classify such scholars as "in the field of humanities and social sciences", in fact, their research projects are concerning, say, "The human impact of the epidemic" and "cultural reasons for differences in national responses". These types of researches

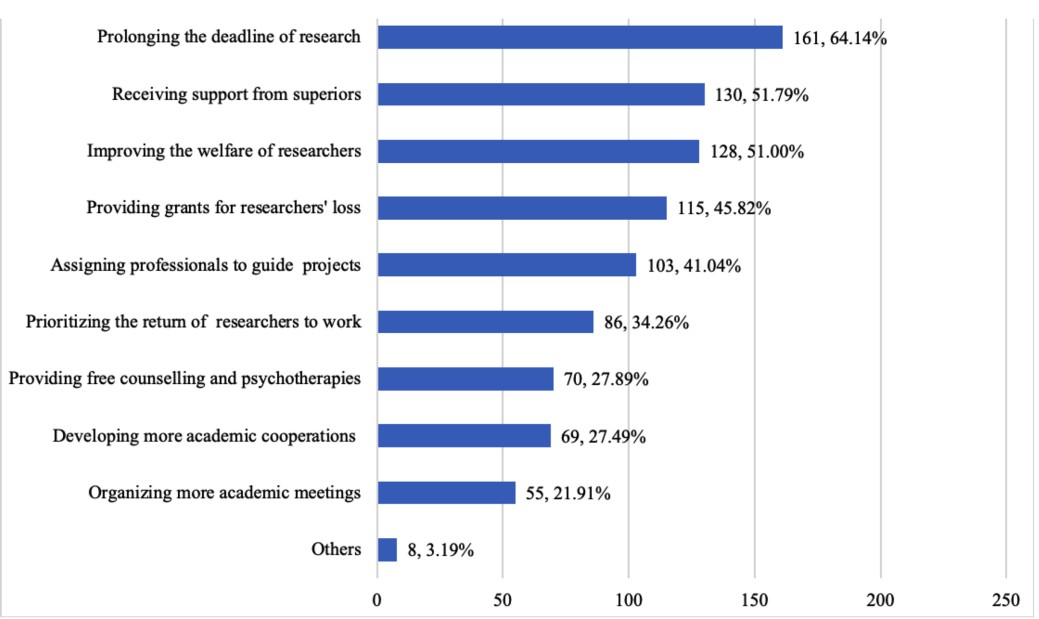

**Figure 1 Some recommendations that researchers considered effective to relieve pressure from the coronavirus pandemic (251 participants).**

**Table 7 A comparison of recommendations endorsed by basic medical researchers and clinical medical staff.**

| Appeals | | Education background | | Total | $\chi^2$ | $p$ |
|---|---|---|---|---|---|---|
| | | Basic research staff (including postdoctoral) | Clinical medical staff (including postdoctoral) | | | |
| Prolonging the graduation/project conclusion/ the deadline of the application of the funds | Disagree | 12 (38.71) | 23 (37.70) | 35 (38.04) | 0.009 | 0.925 |
| | Agree | 19 (61.29) | 38 (62.30) | 57 (61.96) | | |
| Providing funds for the loss of the researches | Disagree | 14 (45.16) | 33 (54.10) | 47 (51.09) | 0.657 | 0.418 |
| | Agree | 17 (54.84) | 28 (45.90) | 45 (48.91) | | |
| Improving the welfare of the researchers | Disagree | 11 (35.48) | 30 (49.18) | 41 (44.57) | 1.561 | 0.212 |
| | Agree | 20 (64.52) | 31 (50.82) | 51 (55.43) | | |
| Providing free consulting and psychotherapies | Disagree | 21 (67.74) | 43 (70.49) | 64 (69.57) | 0.073 | 0.786 |
| | Agree | 10 (32.26) | 18 (29.51) | 28 (30.43) | | |
| Assigning professionals to guide projects | Disagree | 25 (80.65) | 32 (52.46) | 57 (61.96) | 6.928 | 0.008** |
| | Agree | 6 (19.35) | 29 (47.54) | 35 (38.04) | | |
| Encouragement from the superiors | Disagree | 15 (48.39) | 29 (47.54) | 44 (47.83) | 0.006 | 0.939 |
| | Agree | 16 (51.61) | 32 (52.46) | 48 (52.17) | | |
| Prioritizing the return of researches to work | Disagree | 15 (48.39) | 44 (72.13) | 59 (64.13) | 5.038 | 0.025* |
| | Agree | 16 (51.61) | 17 (27.87) | 33 (35.87) | | |
| Organizing more academic meetings | Disagree | 22 (70.97) | 48 (78.69) | 70 (76.09) | 0.673 | 0.412 |
| | Agree | 9 (29.03) | 13 (21.31) | 22 (23.91) | | |
| Develop more academic cooperative projects | Disagree | 23 (74.19) | 39 (63.93) | 62 (67.39) | 0.984 | 0.321 |
| | Agree | 8 (25.81) | 22 (36.07) | 30 (32.61) | | |
| Total | | 31 | 61 | 92 | | |

**Notes:**
* $p < 0.05$.
** $p < 0.01$.

**Table 8 A comparison of recommendations endorsed by masters and PhD students.**

| Appeals | | Education background | | Total | χ² | p |
|---|---|---|---|---|---|---|
| | | Master | PhD | | | |
| Prolonging the graduation or the expected finish time of the experiment | Disagree | 14 (37.84) | 17 (39.53) | 31(38.75) | 0.024 | 0.877 |
| | Agree | 23 (62.16) | 26 (60.47) | 49 (61.25) | | |
| Providing supports for the loss of the researches | Disagree | 19 (51.35) | 23 (53.49) | 42 (52.50) | 0.036 | 0.849 |
| | Agree | 18 (48.65) | 20 (46.51) | 38 (47.50) | | |
| Improving the welfare of the researchers | Disagree | 18 (48.65) | 15 (34.88) | 33 (41.25) | 1.555 | 0.212 |
| | Agree | 19 (51.35) | 28 (65.12) | 47 (58.75) | | |
| Providing free counselling and psychotherapies | Disagree | 27 (72.97) | 31 (72.09) | 58 (72.50) | 0.008 | 0.93 |
| | Agree | 10 (27.03) | 12 (27.91) | 22 (27.50) | | |
| Assigning professionals to guide projects | Disagree | 21 (56.76) | 18 (41.86) | 39 (48.75) | 1.766 | 0.184 |
| | Agree | 16 (43.24) | 25 (58.14) | 41 (51.25) | | |
| Supports from the superiors | Disagree | 14 (37.84) | 20 (46.51) | 34 (42.50) | 0.612 | 0.434 |
| | Agree | 23 (62.16) | 23 (53.49) | 46 (57.50) | | |
| Prioritizing the return of researchers to work | Disagree | 29 (78.38) | 21 (48.84) | 50 (62.50) | 7.405 | 0.007** |
| | Agree | 8 (21.62) | 22 (51.16) | 30 (37.50) | | |
| Organizing more academic meetings | Disagree | 31 (83.78) | 33 (76.74) | 64 (80.00) | 0.616 | 0.433 |
| | Agree | 6 (16.22) | 10 (23.26) | 16 (20.00) | | |
| Develop more academic cooperative projects | Disagree | 30 (81.08) | 33 (76.74) | 63 (78.75) | 0.224 | 0.636 |
| | Agree | 7 (18.92) | 10 (23.26) | 17 (21.25) | | |

Notes:
* $p < 0.05$.
** $p < 0.01$.

**Table 9 Research progress in different fields affected by COVID-19.**

| | | Research field (%) | | Total | χ² | p |
|---|---|---|---|---|---|---|
| | | Science | Humanities | | | |
| Scientific research projects you participated in during the COVID-19 outbreak are | At a standstill | 106 (47.11) | 1 (8.33) | 107 (45.15) | 7.158 | 0.028* |
| | Still under way but at a slower pace than before | 72 (32.00) | 6 (50.00) | 78 (32.91) | | |
| | Completely unaffected | 47 (20.89) | 5 (41.67) | 52 (21.94) | | |
| Total | | 225 | 12 | 237 | | |

Notes:
* $p < 0.05$.
** $p < 0.01$.

are scientific, though they tend to focus more on humanities. Of the 77 professors and lecturers surveyed, 43 (55.84%) indicated that their experiment was at a standstill, while 8 (10.39%) indicated that their experiment was not affected. However, the responses of researchers without professional titles varied, with 43.82% of researchers indicating stagnated experiments and 23.11% indicating unaffected projects respectively (Tables 9 and 10).
**Table 10 Research progress under the influence of COVID-19 by different title of the technical post.**

| | | Title of the technical post (%) | | Total | $\chi^2$ | $p$ |
|---|---|---|---|---|---|---|
| | | Lecturers and Professors | None | | | |
| Scientific research projects you participated in during the COVID-19 outbreak are | At a standstill | 43 (55.84) | 67 (38.51) | 110 (43.82) | 11.453 | 0.003** |
| | Still under way but at a slower pace than before | 26 (33.77) | 57 (32.76) | 83 (33.07) | | |
| | Completely unaffected | 8 (10.39) | 50 (28.74) | 58 (23.11) | | |
| Total | | 77 | 174 | 251 | | |

Notes:
* $p < 0.05$.
** $p < 0.01$.

## DISCUSSION

There may be many subjective or objective factors preventing the achievement of motivating factors like job achievement, income, respect, reputation, work pride, promotion opportunities, etc. Hindered scientific research progress may lead to reduced salaries and promotion opportunities and could delay job achievement. It might also discourage many researchers who had family or other social responsibilities. The resulting stress might be internalized and cause adverse psychological consequences (*Kinman, 2008*; *Liu et al., 2019*). The results showed participants whose progress was affected by the outbreak had higher levels of stress. This is consistent with the study that work interruption is a common source of stress for researchers (*Gmelch, Lovrich & Wilke, 1984*). We found that researchers who reported needing to change their original research programs often faced more pressure. This indicated the change of work content in a short time may be difficult for researchers to deal with (*Kinman, 2001*). Participants who indicated pessimism about halted or slowed research progress also had higher levels of stress than participants who were optimistic. These data provide evidence that we should promote the importance of psychological and mental health in researchers and provide intervention guidance during times such as infectious disease outbreaks (*Jiang et al., 2020*). In addition, previous research has described that most researchers faced unhealthy competition and high levels of competitive pressure at work (*Randall et al., 2019*; *Wellcome, 2020*). In our study, researchers whose colleagues were conducting related research on COVID-19 showed increased stress levels. This suggested that with the full efforts of researchers to study COIVD-19, the stress of scientific research competition also intensified.

To help determine interventions to reduce researchers' stress, we asked researchers to provide suggestions regarding how to respond to the demands. The top recommendation was extending the deadline for experimental projects. This was because the delay of experimental progress often affected original research plans, such as applying for funding and students' graduation. The results showed that "receiving support from superiors" would also help to reduce stress. In previous studies, many respondents reported that their workplace put overwhelming expectations on them and that the superiors' blame led

to an increase in staff dissatisfaction. In contrast, "respect" and "caring for others" were considered positive leadership styles (*Kobulnicky, 1997*; *Merrill, 2015*; *Morsiani, Bagnasco & Sasso, 2017*; *Wellcome, 2020*). In addition, inadequate salary and slow career advancement have been considered as stressors for researchers (*Gmelch, Lovrich & Wilke, 1984*; *Kinman, 2001*). This explains the requirement to improve the welfare of researchers.

Importantly, with graduation deadlines approaching, many students may have felt pressure to complete their qualifications. The lack of science educational activities during the pandemic could entail delay of graduation. Furthermore, perceived stress was correlated with academic level: stress increased with a higher academic level (*Fadhel & Adawi, 2020*). Also, uncertainty around doctoral students and post-doctoral researchers' careers may have made them more vulnerable to publication stress (*Frandsen et al., 2019*). We found that students pursuing a PhD degree showed a stronger willingness to prioritize the return of researchers to work than students pursuing a master's degree.

Researchers in the life sciences and engineering indicated that their scientific research was more severely hindered than those in the social sciences and other fields. Most life sciences and engineering fields rely on experimental facilities to complete their research; the closure of those facilities during the epidemic created a great obstacle for completing their research (*Service, 2020*; *Tencent, 2020a*, *2020b*). In contrast, researchers in social sciences and other fields could often still conduct research activities during the outbreak.

This study has several limitations. The sample size was too small to conduct the population-level sample comparisons that we had anticipated. Further, because this study took place one month after the outbreak began, psychological stress may not have occurred yet. Long-term psychological impacts of infectious disease outbreaks on scientific researchers, such as PTSD, should be investigated in future studies. Finally, we did not compare researcher stress between Hubei (the initial and severe outbreak location) and other regions because there were few respondents from Hubei.

## CONCLUSIONS

Research progress was hindered by the COVID-19 outbreak, especially for researchers in the life sciences (e.g., basic medicine and clinical medicine). Researchers who were affected by the outbreak indicated higher psychological stress levels, especially emotional states, somatic responses and behaviors. Our investigation suggests that the pressure placed on researchers during an epidemic comes mainly from lack of experimental progress and competition among peers. Additionally, clinical medicine researchers were also concerned that the value of their experimental results would be reduced because of delays in progress. The majority of respondents indicated that effective ways to relieve stress included extending deadlines, receiving research support from superiors, and increasing benefits for researchers. The results of this investigation suggest that in addition to focusing on restoring normal order of the laboratory after the novel coronavirus pandemic, it is also important to improve the psychological state of researchers.

### Funding

This study was supported by grants from the National Natural Science Foundation of China (81700658 and 81970248), the Hunan Provincial Natural Science Foundation-Outstanding Youth Foundation (2020JJ3058), and the Medjaden Academy & Research Foundation for Young Scientists (nCoV_MJA20200221). The funders had no role in study design, data collection and analysis, decision to publish, or preparation of the manuscript.

### Grant Disclosures

The following grant information was disclosed by the authors:
National Natural Science Foundation of China: 81700658 and 81970248.
Hunan Provincial Natural Science Foundation-Outstanding Youth Foundation: 2020JJ3058.
Medjaden Academy and Research Foundation for Young Scientists: nCoV_MJA20200221.

### Competing Interests

The authors declare that they have no competing interests.

### Author Contributions

- Xueyan Zhang performed the experiments, prepared figures and/or tables, and approved the final draft.
- Xinyu Li performed the experiments, prepared figures and/or tables, and approved the final draft.
- Zhenxin Liao performed the experiments, analyzed the data, prepared figures and/or tables, and approved the final draft.
- Mingyi Zhao conceived and designed the experiments, authored or reviewed drafts of the paper, and approved the final draft.
- Quan Zhuang conceived and designed the experiments, authored or reviewed drafts of the paper, and approved the final draft.

### Human Ethics

The following information was supplied relating to ethical approvals (i.e., approving body and any reference numbers):

The 3rd Xiangya Hospital Ethics Committee approved this study (20005-IRB).

### Data Availability

Raw data is available as a Supplemental File.

### Supplemental Information

Supplemental information for this article can be found online at http://dx.doi.org/10.7717/peerj.9497#supplemental-information.

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
