# Peer review of "Evaluation of psychological stress in scientific researchers during the 2019–2020 COVID-19 outbreak in China"

_PeerJ, doi:10.7717/peerj.9497_

## Round 0.1 · original submission · Major Revisions

I have read your paper with interest, and received comments from two reviewers. On that basis my decision is to recommend major revisions.

I would like to start by expressing my sympathy for you and your colleagues trying to do research in these very difficult circumstances. The difficulties your survey respondents describe are similar to those we are now experiencing in the UK, and so the topic of your research is very relevant for all of us. Nevertheless, I concur with reviewer 1 who queries whether this paper requires rapid review. Given that the pandemic seems likely to last for some months, it seems unlikely that much can be done to act on the recommendations right now, but the findings could be useful once the pandemic is largely over, at which point there will be scope to take actions to redress the problems.

The comments from reviewers are in good agreement with one another, and are constructive in suggesting how to improve the paper, so please do look at their detailed comments and address these if you plan to revise the paper.

Both reviewers picked up on several of the same points regarding things that could be improved upon in the paper.
1. Both have noted places where the English is unclear. I realise that it must be challenging writing in English and I am pleased to see that the reviewers have made some explicit suggestions for rewording that I hope you can incorporate. Reviewer 2 has included an annotated version of the pdf (nb the report describes this as an 'annoyed' version. I assume this is an autocorrect bug!).
2. Both note the need to report the questionnaire items.
3. There is a need to relate this study to other relevant literature (see suggestions by reviewers).
4. Make a distinction between hypotheses specified in advance, and more exploratory analyses.
5. Explain how the sample was recruited
6. The reviewers also suggest alternative approaches to reporting the data that would be clearer.
7. Both reviewers felt the analysis was not optimal. Please see their suggestions for alternative approaches.

Two additional points:

The current title of the paper is not appropriate, because the paper is about 'psychological' rather than 'physiological' stress.

Thanks for uploading a copy of the ethics approval document. This was rather confusing because it made reference to a 'clinical trial' and genetic data, neither of which applies to this study. It was also dated after the data collection. I wondered whether you are required in China to obtain ethical approval for this kind of survey study, as this is not the case for all countries. Indeed, I see no ethical issues with the research. If this is the case it would be worth just noting this, e.g. by saying 'It is not required by our institution to obtain ethical approval for a survey with a nonclinical sample and anonymised data, but we did obtain retrospective approval for this study.'

·

Basic reporting

The manuscript generally is clear but there are some lapses in the English. Occasionally these make it hard to work out the intended meaning. This needs to be tightened up but mostly this cosmetic (annoyed pdf attached to this review).

The literature is a little sparse and confined to recent work. There is however older work on stress in academia and among researchers. There are also some recent work that is probably relevant:

Some suggested additional context:

https://wellcome.ac.uk/sites/default/files/what-researchers-think-about-the-culture-they-work-in.pdf

https://www.thelancet.com/journals/lancet/article/PIIS0140-6736(20)30460-8/fulltext

https://www.tandfonline.com/doi/abs/10.1080/01443410120090849

https://www.tandfonline.com/doi/abs/10.1080/0144341960160104

Overall the structure is mostly fine, but I think the specific hypotheses and links to the analyses reported lack clarity. To the extend that there is a confirmatory element to the research I'd like clear hypotheses stated and linked to specific analyses. There is also clearly an exploratory element and I'd like to see that facilitated by reporting: a table of means and SDs of the measures and a correlation between the key variables. I'll go through analysis in turn and make some suggestions for improving clarity of reporting below.

Experimental design

The sampling strategy is not adequately described. This needs to be stated (i.e., how were email addresses targeted). I'm not sure how participants were randomly distributed - so I suspect this was an opportunity sample. Even if randomly distributed the responses are not random so the sample is probably best described as an opportunity sample.

As stated earlier the primary research question and hypotheses need to be clearer.

Making the survey Qs available to readers (if necessary stripping out copyrighted scales) would be helpful.

Validity of the findings

The data are provided and look reasonable - however I have not reproduced any analyses.

The analyses are a bit confusing in places - partly because I don't know what the main research Qs are. I'll go through the main analyses table by table.

To interpret these it would help to have the means and SDs (maybe skew and kurtosis) and in the scale descriptions the min and max of the scales should be reported. It would also be useful to report internal consistency reliability for the scales (coefficient omega or coefficient alpha - especially the scales that are modified.

Table 2a. The reporting is a little non-standard but is equivalent to separate analyses of the four outcomes with whether the researchers agree that research been delayed as a predictor. I'd consider correcting for multiple testing (though as it happens a Hochberg adjustment gives the same pattern of statistical significance).

Table 2b. I don't think this is a sensible way to analyse the data. First some categories are too small (e.g., the relaxed category has n = 1 and should be collapsed into one of the others). Second the categories aren't obviously ordered and seem to mix outcome with predictor variables (i.e., whether the project was impacted and the state of the researcher). I'm not sure what is the best way to sort this out. Possibly there is a clear ordering here and it could be handled more neatly by correlating the ordinal response with the continuous outcome. I think a Spearman or Pearson correlation would suffice here as there is a single predictor.

Table 3. I'm not a huge fan of analysing subgroups like this. It is easy to detect spurious patterns via multiple testing. It is also described in the table as a stepwise regression. Stepwise regression biases tests and estimates. I'd recommend a single regression with simultaneous entry of predictors. I'd do this in stages with demographics entered as a block and then the predictors of interest. Instead of subgroup analyses include large subgroups as dummy coded predictors and include interactions with key predictors.

Table 4. These sorts of tables are hard to interpret. Collapse the small categories into similar groups (all PhD students for instance). To aid detection of patterns I'd consider looking at the standardised residuals or adjusted standardised residuals (the former are a bit conservative - but that tends to make them useful in situations like this as it is less important to correct for multiple testing). The standardised residuals have an approximate z distribution and thus if exceed +/- 1.96 are statistically significant at p < .05.

Table 5a. Collapse social science and others (small n and similar patterns). More importantly I think the irrelevant category needs to be excluded from the analysis in the bottom analysis (mention this in the notes to the table). Including it makes no sense.

Table 5b. Here and the top analysis in table 5a you have an ordinal outcome - however the title variable, while arguably ordinal, is showing a non-monotonic pattern. There is a case then I think for an ordinal regression with research progress as an outcome and title dummy coded. This would allow other predictors to be included - but that might not be sensible if you have no specific hypotheses about them. Gender might be worth including. So I can see case for keeping the original chi-square analysis (and maybe also looking at the standardised residuals).

Additional comments

This is a potentially useful data set that could be important in highlighting how the present pandemic is impacting on research. The sampling strategy, research question and analyses could be better described and the presentation strengthened. The literature is also focused a bit narrowly on recent work without linking to prior research on researcher burn-out or stress (which would probably provide additional support for their main conclusions).

Although the research questions need to be clarified and and specific hypotheses flagged I'd also ask the authors to highlight the largely exploratory nature of the research where appropriate.

·

Basic reporting

1. The language used could be clearer throughout the manuscript. I note several instances in my comments below. In some cases it is merely that very long sentences could be split into separate ones, e.g. “Due to the severe epidemic situation and in order to avoid further transmission, most industries had been forced to shut down temporarily and science and education activities were paused in China, which had caused people much inconvenience(ScienceMag.org 2020b).” (page 6). Similarly, the following sentence is very hard to follow “Animal centers and practical labs were borne the brunt not to be admitted, and many scientific and social congresses and symposiums were cancelled, and postgraduates and scientific workers were confined to be back to their workplaces”. In particular the “confined to be back to their workplaces” is very hard to follow.
2. There are instances of unnecessary repetition throughout the manuscript. E.g. the 23 questions of the perceived stress scale. Stating this twice made the section difficult to follow – at first I thought these were different scales. Please revise the wording for clarity
3. While I appreciate more colloquial language in manuscripts, this manuscript at times does not read like a research article. E.g. “it can be roughly seen from this question…” The writing should be revised to be clearer and more precise.
4. Please report very small p values as < .001, rather than p=.000

In sum, I do not feel that this manuscript meets the standards of the journal in this section.

Experimental design

My comments on the experimental design largely fall into the general comments under "methods"
1. Given that the measures used are novel, all items should be provided in the report, e.g. as an appendix. It came as a complete surprise that one of the possible responses was “about to collapse”. As the measures are adapted/new it is particularly important that the reader is provided with this information. When checking the raw data I noticed items that were not discussed in the manuscript, perhaps ones that were removed in the stepwise regression analyses. I suggest that a table is included with summary statistics for all items in the surveys – perhaps merely a summary score for the standardised measure.
2. Please include details on how the questionnaire randomly distributed. The authors should provide more information on this and the recruitment process.
3. How did the authors check that respondents understood the meaning of the question?

Much work would be needed for this manuscript to meet the journal standards for this section

Validity of the findings

1. As exemplified in specific comments in the general comments header, the paper would be clearer if results and speculation of the results were more firmly distinguished throughout the manuscript. Similarly, the authors should be careful to not make inferences of the results that the analyses did not test, e.g. specific differences between two groups in the Chi Squared tests.

Additional comments

The authors present a survey study asking what impact the COVID-19 pandemic (and social isolation measures) has on researchers work capacity mental health. The abstract supports many of the shared feelings among researchers during the COVID-19 outbreak. I was excited to read a paper examining the impact of research disruptions on researcher mental health; we need data on this in order to understand how to best support researchers. Even simple survey statistics – as I expected from this paper – would be useful to use in future appeals to improving working conditions and support for researchers.

I empathise with the intentions of this research and sympathise with the author’s desire to have their work published. But, I am not clear what this paper offers above e.g. the blogpost referenced throughout the paper (Sciencemag.org, 2020a). The authors comment reads “This is a clinical investigation of COVID-19 epidemic, which will help others around the world during this outbreak. We hope you could consider it for a rapid reviewing and editing process. Many thanks.” This gave me some enthusiasm for the paper. Yet, the paper is not a clinical investigation, nor do I feel the results would help others during the outbreak. The authors received rapid review for this paper, yet I cannot see why this paper needs urgent publication. I cannot help but think that other papers would be more deserving of reviewer efforts at this time.

Core to my comments below is that this paper requires substantial revision and proof-reading to be able to review in its entirety.

Abstract:
1. I suggest stating “Chinese government” for additional clarity
2. The sentence beginning “the pressure on colleagues…” reads oddly with the word “researches”.
3. I am not clear what is meant by “a great influence on the experiment” in the final sentence.

Introduction:
4. The abbreviation “COVID-19” is introduced twice in the first paragraph. No need for the second.
5. The majority of the literature discussed in this manuscript are papers from 2020 focused on COVID-19. Much work has been already conducted to investigate researcher and student mental health (e.g. to name two https://wellcome.ac.uk/reports/what-researchers-think-about-research-culture and https://www.nature.com/articles/d41586-019-03459-7) in relation to the issues raised in this paper. This limits some of my enthusiasm for this study as it does not situate it within known issues within researcher and student mental health, including work-related stresses. So, while I am concerned with the same issues raised by this paper the introduction will need to be reworked to situate the paper within the already existing literature.
6. In the final paragraph, the authors could describe the present study with more clarity. E.g. what diverse populations? How was the study conducted? As it stands, most of the paragraph actually appears to discuss the results and indicate points in the discussion section.

Methods:
Note: Other comments appear under the “experimental design” subheading above.
7. The questionnaires section also somewhat describes the analysis plan, making it difficult to follow. I suggest reconfiguring the methods section to more clearly differentiate these aspects.

Results:
8. The results section would be clearer if discussion of the results were moved to the discussion section. E.g. most of the ‘basic information of the participants’ paragraph is discussion related, and the sample summary is already in table 1.
9. The results section reads more as a qualitative analysis, rather than a quantitative analysis of data collected. I suggest revising the results to more specifically describe the statistics performed, and move any discussion of implications/speculation to the discussion section.
10. At first glance, I was concerned that the authors were selectively reporting significant results in the regression analyses. Then table 3 made it clear that a stepwise approach was used. First, relevant details such as this should be presented in the data analysis description. Secondly, there are several strong arguments against a stepwise approach (e.g. https://journalofbigdata.springeropen.com/articles/10.1186/s40537-018-0143-6 and https://towardsdatascience.com/stopping-stepwise-why-stepwise-selection-is-bad-and-what-you-should-use-instead-90818b3f52df). I suggest reporting all predictors, and report any corrections for multiple comparisons applied, or not.
11. The description of the three researcher appeals is hard to follow. Several appeals are discussed in the same sentence. The wording suggests that groups differed in their rate of appeal. Yet, none of the Chi squared tests would be significant if multiple comparisons were considered. Nor were tests conducted between individual groups.
12. In the section “The research progress affected by COVID-19 differs from researcher’s identities”: It would be helpful to know what analyses were conducted before saying that there were statistical differences. This would allow the reader to understand the meaning of the analyses conducted. The statistics reported do not correspond to the description of results. The Chi squared test does not tell us that one individual group scores differently to another specific group, additional tests would be needed to address this.
13. The authors reference tables 2a and 2b at the end of the results section, I assume that they mean 5a and 5b?

Discussion:
14. As mentioned earlier on the discussion section, the literature discussed here is limited. For this work to have more utility it must be situated well in the existing literature – this would provide much more support to
15. An aspect that makes the manuscript difficult to follow throughout is a lack of clarity between discussion of results and implications. Discussion of results is intertwined with potential implications that were not examined within the survey and analysis.
16. The limitations section exemplifies my previous two points. No literature is referenced to support the statements, and it is not entirely clear what the suggestions would add other than a cursory observation that other work could be done.
17. Additionally, the authors should add more detail to the statement about power of this study. Was a power analysis conducted? Why not collect data for longer if this is the case? What is meant by underpowered in this case?

Tables and Figures:
18. Please indicate what the +/- indicates in table 2 – I assume standard deviation?
19. Please also note what the numbers in parentheses are in table 4.

Data:
20. I want to thank the authors for providing the raw data - this is something I stand firmly behind. Could the authors provide a copy of the raw data with the numeric values instead of the worded item responses?
21. Perhaps the authors would be willing to also share their analyses syntax? Of course this is not a condition of publication in PeerJ, but it would be a nice example of open practices.

In the interest of openness, I always sign my reviews,
Sam Parsons

---

## Round 0.2 · Major Revisions

I have read your revision carefully and also received comments from one of the original reviewers. Because of difficulties getting reviewers at this time, I am providing a more detailed review than would be usual for an editor. My decision is that although the manuscript is much improved, a close reading of this new version reveals many problems and places where the study is unclear. I will give detailed feedback, in the hope these issues can be addressed, but I'm afraid a fair amount of revision would be needed before it can be considered for publication.

Please note I suggest adding 2019-2020 to the title. Sadly, we cannot assume there will just be one pandemic, so we need to be specific about dates.

Major issues

One point of lack of clarity is right at the end of the paper where you introduced the term NCP, defined as 'novel coronavirus pneumonia'. This was confusing because up to this point my impression was that you had been talking more about effects of being in lockdown and unable to work, rather than effects of physical illness. You used the term NCP also in Table 3 , where you say 'the NCP delays the completion date...'. Should this refer to 'pandemic' rather than 'pneumonia'? We need to be clear whether aspects of your paper are only focussed on those who have experienced pneumonia – or whether, as I had thought originally, you include those who have not fallen sick, but who have been nevertheless unable to work. The NCP term is also used in Table 4.

In the text it is stated that 'Two similar questions were separately set in the questionnaire, and the validity of the questionnaire was judged by comparing the consistency of the respondents' answers'. I was unclear which items this referred to, and I could not find any further mention of this validity check. Please be more specific.

The nature of the 'appeals' is rather unclear. I had thought at first that these were gathered from free text responses, but now the questionnaire is provided, it is clear these are options that the respondents could select from. I have suggested rewording accordingly in the attached word document.

In Table 3, you analyse the impact of the delay of project on stress levels. It is unclear why you picked specifically the delay to research projects as the independent variable for this comparison. Interestingly, when you look at Table 4, this variable does not emerge from the multiple regression as a significant independent predictor. But that is perhaps not surprising because the multiple regression will indicate which variables are significant *after taking all other variables in the equation into account*. I'm not convinced that this is a particularly helpful way to look at the data, given that your questionnaire items are likely to be correlated. Having inspected your data, I think a more descriptive account would make better sense. You could either just present a set of t-tests on all the variables from table 4, comparing overall stress in relation to the binary responses, and adopt a comparison for multiple comparisons, such as Bonferroni. Or you could attempt some data reduction, to create fewer categories from the 14 variables, e.g. by using principal component analysis. I'm sorry to make this new suggestion at this point, but it was only when I was able to scrutinise the results in table 4 that I realised they were misleading, particularly in terms of the apparent inconsistency with table 3.

In either case, we need to have the correlation matrix for all the variables that were in the regression, with means and SDs as additional rows at the bottom. This could be included as an appendix. It allows readers to see how independent the data is for the different items.

In Tables 5 and 6, it looks again as if you have just selected out those comparisons that yield p < .05, but if you analysed all variables, then these should be reported, and an appropriate correction for multiple comparisons should be made.


Minor wording changes
For modifications needed in the main manuscript, please see the attached document with tracked changes: this incorporates minor modifications to wording both by me and by the reviewer, and you should accept these in your revision. In addition, please clarify why the Tytherleigh reference has a * in the name.

The tables and figures also need some changes, as noted here:

Figure 1: As noted by the reviewer, 'appeals' seems the wrong word here, and I have suggested you alter this to 'recommendations' throughout including in the title to this figure. Change 'researches' to 'researchers'
Please correct spelling/change wording in the Y-axis legend as follows:

encouragements -> support
profesionals -> professionals
conceling -> counselling

Table 1
rename title to 'Demographic characteristics of respondents'

the phrase '985,211 universities' is not widely known outside China. I looked on Google and was still rather unclear. Is there another phrase that you could use here? Do these correspond to 'research-intensive universities'.

Table 2: It was unclear which items belonged with which scale. Please specify the questionnaire item numbers used for each scale in the table.

Table 3
The title is unclear. This was in part because of use of NCP here. My impression is that you are reporting differences between researchers who have and have not experienced delay in the completion date of their scientific research project.
If so, then the header should read 'Completion date of scientific research project was delayed by coronavirus.' Or are you specifically talking about cases where researchers experienced delay because they or colleagues had pneumonia?

Table 4
NB As noted above, I think simple comparisons of those responding 1 vs 0 would be easier to interpret than the multiple regression.
In terms of row labels there are many errors:
row 1: NCP – see above re this term; are we only talking about cases where people had pneumonia? If not, reword to "Delayed completion date of research project"
Row 2: reword "Blocked research project affects timing of graduation/project completion/funding application"
Row 3: "Original research program needs to be changed or cut"
Row 4. – I did not understand what this meant! Please give a clearer term than "Epidemic resistance reduces the value of research"
Row 5. – again this was hard to understand – what is meant by "failure of original experimental results"
Row 6 – Need for more input of energy by researchers (? this is a bit unclear)
Row 7 – Actual or potential great loss for projects
Row 8 – Impact on academic exchange activities
Row 9 – impact on cooperation with other organisations
Row 10 – 'Colleagues have carried out research projects on NCP' – this is very confusing. Does it refer to concerns that colleagues may be infectious, and put the researcher's health at risk; or is the concern that colleagues may have an advantage because they are able to do research on a relevant topic, whilst the respondent cannot do so?
Row 11 – 'Launch projects concening the NCP'. Again, this is very unclear.
Row 12 – 'Pressure on colleagues to carry out research projects on NCP' – who is exerting the pressure here?
Row 13 – reword 'Disruption of scientific research adversely affects your career'.
Row 14 – reword 'Disruption of scientific research has caused adverse effects on salaries'

Table 5.
Reword title 'A comparison of recommendations endorsed by basic medical researchers and clinical medical staff'
It looks as if you have just selectively reported those appeals ( ie recommendations) where the two groups differ. You should report the data for all the appeal categories, regardless of whether significant or not.

Table 6
Reword title' A comparison of recommendations endorsed by masters and PhD students.' Correct label 'researches' to 'researchers'
The same point applies here – you should report all the comparisons you conducted – not just those meeting a significance threshold.

Table 7.
These results are rather odd, because you would not expect scholars in humanities to be conducting scientific research. It is rather unclear, therefore, whether the statement that projects are unaffected is because there are no projects. Did you have any way of confirming that these people were actually involved in science projects?

·

Basic reporting

The revised MS is much clearer and the findings are related appropriately to the wider literature on researcher stress. There are still a few areas where the English could be improved - though this is mostly very minor or to make key points clearer. I have edited the word document* with track changes to pick up most of the required changes - with a few queries/comments. In particular the term "appeals" might be easier to understand as "proposed solutions" (throughout the MS including tables).

*It appears I can not upload the word version I have edited - I have converted this to a pdf, but can provide the word version if required.

Experimental design

The survey is explained much more clearly.

Validity of the findings

The conclusions are more measured and better supported by the evidence presented. In particular collapsing categories and reporting slightly simpler analyses has improved the MS.

While it isn't directly relevant to the revisions I note that I didn't explain my recommendation about using standardised residuals of the chi-square analysis clearly. It is no longer relevant to the MS, but this is a paper describing the approach I was suggesting:

MacDonald PL, Gardner RC. Type I Error Rate Comparisons of Post Hoc Procedures for I j Chi-Square Tables. Educational and Psychological Measurement. 2000;60(5):735–754.

Additional comments

I am recommending minor revisions only to tidy up the minor issues with the MS that remain. (Apologies for the autocorrect error on my previous review - it was indeed "annotated" rather than "annoyed".)

---

## Round 0.3 · Major Revisions

Thank you for the revised manuscript.
I'm sorry but I am now confused about how 'total stress' score is computed. I had assumed this was the total of the scores on emotional state, somatic responses, sleep quality and behaviour, but the totals in the new tables 4, 5 and 6 are much higher than this. I cannot make an editorial decision on this manuscript unless this is clarified.

---

## Round 0.4 · accepted · Accept

Thank you for the clarification regarding the total stress measure.
It is fortunate that we caught this prior to publication.

I am pleased to say that I now find your article acceptable for publication in PeerJ, provided you can just make some very minor amendments to aspects of the text as follows:

Line 81-85, reword to
In addition, the respondents' answers were internally consistent. In particular, the 193 respondents who chose 'At a standstill' or 'still in progress but slower than before' due to the epidemic situation (question 37) responded with psychological states (questions 38-39) that matched this answer.

Line 141, 'peer pressure that their colleagues have...' reword to 'peer pressure because their colleagues had...'

Figure 1: legend needs correcting. The sentence is repeated and also has typos in it. It should read:
'Some recommendations that researchers considered effective to relieve pressure from the coronavirus pandemic (251 participants).'

Table 4. Reword 'constant' to 'variable'
The item 'Launch projects concerning the NCP' is still totally unclear to me.
Please can you rephrase this to make it clearer what the respondents are judging here.

Table 6. I am still a bit uncertain here, but I think you mean to say that the presence of colleagues working on COVID-19 causes stress on researchers, in which case this should read:

"The impact of 'Feeling great pressure from colleagues who conduct projects on COVID-19' on stress levels"

assuming this is the correct interpretation, please be careful also to correct this in Table 4.